# Direction-of-Arrival Estimation for a Random Sparse Linear Array Based on a Graph Neural Network

**DOI:** 10.3390/s24010091

**Published:** 2023-12-23

**Authors:** Yiye Yang, Miao Zhang, Shihua Peng, Mingkun Ye, Yixiong Zhang

**Affiliations:** 1School of Electronic Science and Technology, Xiamen University, Xiamen 361005, China; 36620211150369@stu.xmu.edu.cn (Y.Y.); 36620221150427@stu.xmu.edu.cn (S.P.); zyx@xmu.edu.cn (Y.Z.); 2Tung Thih Electron Research Center, Xiamen 361021, China; yemkun0720@gmail.com

**Keywords:** direction-of-arrival (DOA) estimation, sparse linear array, graph neural network, single snapshot, array signal processing

## Abstract

This article proposes a direction-of-arrival (DOA) estimation algorithm for a random sparse linear array based on a novel graph neural network (GNN). Unlike convolutional layers and fully connected layers, which do not interact well with information between different antennas, the GNN model can adapt to the goniometry problem of non-uniform random sparse linear arrays without any prior information by applying neighbor nodes’ aggregation and update operations. This helps the model in learning signal features under complex environmental conditions. We train the model in an end-to-end way to reduce the complexity of the network. Experiments are conducted on the uniform and sparse linear arrays for various signal-to-noise ratio (SNR) and numbers of snapshots for comparison. We prove that the GNN model has superior angle estimation performance on arrays with large sparsity that cannot be used by traditional algorithms and surpasses existing deep learning models based on convolutional or fully connected structures. The proposed algorithm shows excellent DOA estimation performance under the complex conditions of limited snapshots, low signal-to-noise ratio, and large array sparsity as well. In addition, the algorithm has a low time calculation cost and is suitable for scenarios that require low latency.

## 1. Introduction

Direction of arrival (DOA), which realizes the spatial localization of signal sources by analyzing the signals received from array antennas, has always been a key technology in array signal processing. It has been widely applied in many scenarios, including airborne radar [1,2], underwater surveying [3,4], medical diagnostics [5], and autonomous driving [6,7]. The research challenges of DOA estimation are mainly focused on high-precision estimation under the conditions of limited snapshots, low signal-to-noise ratio (SNR), and sparse arrays.

Traditional DOA estimation algorithms can be divided into three categories. The first one comprises beamforming algorithms, such as digital beamforming (DBF) [8], minimum variance without distortion response (MVDR) [9], etc. The second one includes super-resolution goniometry algorithms based on subspace, such as estimating signal parameters via rotational invariance techniques (ESPRIT) [10], multi-signal classification (MUSIC) [11], and some of its derivatives [12,13,14]. The third one comprises sparse reconstruction algorithms, such as atomic norm minimization [15], sparse Bayesian learning (SBL) [16], compressed sensing (CS) [17,18] algorithms, etc. These algorithms rely on traditional array structures to achieve accurate DOA estimation with high SNR and many snapshots. However, non-ideal conditions such as limited snapshots and array defects lead to the degradation of traditional algorithm performance, and calculations such as feature value decomposition and spectral peak search are difficult to adapt to the real-time requirements of the algorithm in engineering.

In order to solve this problem, many scholars have introduced neural networks into the field of DOA estimation [19,20,21,22,23] and have achieved angle estimation by learning the nonlinear relationship between the output of the array and the position of the spatial signal source. Deep learning has been adopted for DOA estimation in large-scale MIMO arrays [24]. The literature shows that the uniform array goniometry algorithm based on a convolutional neural network has a significantly better DOA estimation accuracy than the traditional super-resolution estimation algorithm under the conditions of low SNR and a small number of snapshots. Furthermore, a DOA estimation algorithm based on a convolutional recurrent neural network (CRNN) [25] is proposed and can produce high-precision DOA estimation results under different SNR. The denoising autoencoder (DAE) neural network structure [26] is added to the literature, and the DOA detection accuracy is higher than that of the basic autoencoder by recovering the feature information. Recently, the literature has combined the sum-difference array with deep neural networks [21], demonstrating that neural networks can better adapt to array defects.

These studies show that DOA estimators based on neural networks have a faster computation speed and more robust estimation effects than traditional algorithms and can adapt to more complex environments and interference situations. However, most of them focus on uniform linear arrays with more than 200 snapshots and high SNR of 10 dB or more, which are barely implemented in reality. In addition, fewer DOA estimation studies on sparse arrays with random missing elements have been developed.

The CNN and MLP structure cannot make good information communication between antennas, and it is difficult to adapt to situations in which an array has large sparseness. However, a graph neural network (GNN) [27] is a neural network algorithm with nodes and adjacency matrices, which can well update the graph structure with information between neighbor nodes.

In this work, we propose a novel DOA estimation algorithm based on a graph neural network (GNN) within the GraphSAGE [28] structure to achieve high-precision DOA estimation in complex environments, such as those with low-SNR states, low numbers of snapshots, defective arrays or a combination of these situations. Since non-uniform random sparse arrays have missing elements, GNN can update the feature information of each element by aggregating the received signals between neighbor elements and can restore, to some extent, element information that is randomly missing in the non-uniform array. Hence, the proposed DOA estimation algorithm based on a GNN can achieve strong robustness.

The rest of this article is organized as follows: Section 2 describes the model of array signal and the formation and feature generation process for random sparse arrays. Furthermore, the same section proposes a DOA estimation framework based on a GNN and clarifies how the input and output meet the DOA estimation requirements, and it also introduces the training strategy of the neural network. Section 3 sets the simulation parameters and conducts experimental simulation to verify that the proposed algorithm is superior to previous algorithms based on neural networks and traditional algorithms adopted in random sparse arrays under the conditions of a small number of snapshots or even a single snapshot and low SNR. Section 4 concludes this paper.

*Notations*: ( · )*, ( · )*^T^* and ( · )*^H^* represent the conjugate operator, transpose operator and conjugate transpose operator of the matrix, respectively. E( · ) indicates the expected operator and ${\left\|\cdot \right\|}_{2}$ represents the ***L_2_*** norm operator of a matrix. *Re*( · ) and *Im*( · ) represent the real and imaginary parts of a complex value, and $j=\sqrt{-1}$ represents the imaginary unit. C and R represent the fields of real numbers and complex numbers.

## 2. Materials and Methods

### 2.1. Signal Model

#### 2.1.1. Uniform Linear Array Signal

As an assumption, *K* independent far-field narrowband signals are incident on an *M*-element uniform linear array. As summarized in Figure 1, the uniform element spacing *d* is half-wavelength $\lambda_{0}$ in free space. The incident angles of signals are $\theta_{1},\theta_{2},\ldots ,\theta_{k}$, and the received signals of a ULA are calculated as follows [29]:(1)$$x_{m}\left(t\right)=\sum_{k=1}^{K}a\left(\theta_{k}\right)s_{k}\left(t\right)+n_{m}\left(t\right),for\;m=1,2,\ldots ,M,$$
where $a\left(\theta_{k}\right)$ represents the direction vector of $\theta_{k}$; $s_{k}(t)$ represents the envelope of the *k*-th signal; and $n_{m}(t)$ represents the white Gaussian noise generated at the *m*-th element with a mean of zero and a variance of noise power $\sigma^{2}$. The array output signal is sampled at moment *t*, and the received signal can be expressed as a matrix.
(2)$$X(t)={\left[x_{0}(t),x_{1}\left(t\right),\ldots ,x_{M-1}(t)\right]}^{T}=A\left(\theta \right)S\left(t\right)+N(t),$$
where
(3)$${S(t)=[s_{1}\left(t\right),\;s_{2}\left(t\right),\ldots ,s_{K}(t)]}^{T},$$
(4)$$A\left(\theta \right)={\left[a(\theta_{1}),a(\theta_{2}),\ldots ,a(\theta_{K})\right]}^{T},$$
(5)$${N(t)=[n_{0}\left(t\right),\;n_{1}\left(t\right),\ldots ,n_{M-1}(t)]}^{T},$$
where X(t) denotes the array-received signal matrix; S(t) indicates the signal envelope matrix; and N(t) represents the array noise matrix. $A\left(\theta \right){\in C}^{M\times K}$ is the array manifold matrix, and $a\left(\theta_{k}\right)$ is expressed as follows:(6)$$a\left(\theta_{k}\right)={\left[1,e^{\frac{j2\pi dsin\theta_{k}}{\lambda_{0}}},\ldots ,e^{\frac{j2\pi (M-1)dsin\theta_{k}}{\lambda_{0}}}\right]}^{T}.$$

The covariance matrix $R_{xx}\in C^{M\times M}$ of the array-received signal matrix X(t) can be expressed as follows [30,31]:(7)$$R_{xx}=E\left\{X(t)X^{H}(t)\right\}=A\left(\theta \right)R_{ss}A^{H}\left(\theta \right)+\sigma^{2}I,$$
where $R_{ss}$ represents the covariance matrix of signal S(t), $\sigma^{2}$ is the noise power and ***I***$\in R^{M\times M}$ is the identity matrix.

#### 2.1.2. Random Sparse Linear Array Signal

Random sparse linear arrays with different degrees of sparsity are also illustrated in Figure 1 for comparison. The aperture sizes of RSLAs are kept unchanged to match that of a ULA. Only one randomly generated sparse array for each sparsity rate is provided. The first and last elements of all arrays are reserved. A certain number of missing elements are randomly generated inside the array according to the specified sparsity rate. Hence, the received signals for the missing elements in the RSLA are zero.

As an assumption, the received signal matrix of the RSLA is denoted by $\tilde{X}(t)$. The manifold matrix $\tilde{A}\left(\theta \right)$ of the RSLA can be generated from $A\left(\theta \right)$, only by setting $a(\theta_{K})$ in $A\left(\theta \right)$ at 0 for the missing elements. Thus, $\tilde{X}(t)$ is expressed as follows:(8)$$\tilde{X}(t)=\tilde{A}\left(\theta \right)S\left(t\right)+N(t),$$
where $\tilde{A}\left(\theta \right)={\left[\tilde{a}(\theta_{1}),\tilde{a}(\theta_{2}),\ldots ,\tilde{a}(\theta_{K})\right]}^{T}$, and $\tilde{A}\left(\theta \right)$ is a sparse matrix.

The covariance matrix ${\tilde{R}}_{xx}\in C^{M\times M}$ of the random sparse array receiving signal $\tilde{X}(t)$ is as follows:(9)$${\tilde{R}}_{xx}=E\left\{\tilde{X}(t){\tilde{X}}^{H}(t)\right\}=\tilde{A}\left(\theta \right)R_{ss}{\tilde{A}}^{H}\left(\theta \right)+\sigma^{2}I,$$
where $R_{ss}$ is the covariance matrix of signal S(t), $\sigma^{2}$ is the noise power and *I*$\in R^{M\times M}$ is the identity matrix.

#### 2.1.3. Preprocessing of the Array Signal

The solution for the covariance matrix $R_{xx}$ and ${\tilde{R}}_{xx}$ is obtained under ideal conditions, but in a real system, the original covariance matrix is replaced by a sampling covariance matrix ***R***$\in C^{M\times M}$. In order to facilitate the construction of mathematical models, the received signal ***X***(t) of a uniform linear array and the received signal $\tilde{X}(t)$ of a random sparse array are uniformly defined as $\hat{X}(t)$, so it can be universally applied to the case of uniform arrays and sparse arrays. For the case where the received signal has multiple snapshots, the sampling covariance matrix expression of the array is:(10)$$R=\frac{1}{T}\sum_{n=1}^{T}\hat{X}\left(t_{n}\right){\hat{X}}^{H}\left(t_{n}\right),$$
where T is the number of snapshots, and $\hat{X}(t)={\left[{\hat{x}}_{0}(t),{\hat{x}}_{1}\left(t\right),\ldots ,{\hat{x}}_{M-1}(t)\right]}^{T}$.

Since the covariance matrix R is a complex matrix, complex values cannot be directly used as input features of graph convolutional neural networks, so this work processes the covariance matrix as follows:(11)$$\bar{R}=\left[Re\left(\begin{array}{c} R_{1,1}\;\;\;R_{1,2}\;\;\ldots \;\;R_{1,M}\\ R_{2,1}\;\;R_{2,2}\;\;\ldots \;\;R_{2,M}\\ \;\;\;\;\vdots \;\;\;\;\;\;\;\;\;\vdots \;\;\;\;\;\;\ddots \;\;\;\;\;\vdots \;\;\;\;\\ R_{M,1}\;\;R_{M,2}\;\;\ldots \;\;R_{M,M} \end{array}\right),Im\left(\begin{array}{c} R_{1,1}\;\;\;R_{1,2}\;\;\ldots \;\;R_{1,M}\\ R_{2,1}\;\;R_{2,2}\;\;\ldots \;\;R_{2,M}\\ \;\;\;\;\vdots \;\;\;\;\;\;\;\;\;\vdots \;\;\;\;\;\;\ddots \;\;\;\;\;\vdots \;\;\;\;\\ R_{M,1}\;\;R_{M,2}\;\;\ldots \;\;R_{M,M} \end{array}\right)\right],$$
where $R_{i,j}$(*i* = 1, 2, 3, …, *M*; *j* = 1, 2, 3, …, *M*) represents the row-I and column-j element of the covariance matrix ***R***.

Normalizing the matrix $\bar{R}$ according to Equation (12) the matrix $\hat{R}$ can finally be used as the input feature of the neural network.
(12)$$\hat{R}=\bar{R}/{\left\|\bar{R}\right\|}_{2},$$
where ${\left\|\cdot \right\|}_{2}$ represents the ***L_2_*** norm operator.

### 2.2. DOA Estimation Framework Based on GNN

#### 2.2.1. Graph Convolutional Network

Graph convolutional neural networks have nodes and adjacency matrices. In this study, we introduce a residual graph convolutional neural network based on GraphSAGE to process sparse array signals. The process of working with GraphSAGE consists of two steps: aggregation and update. It is conducted to perform sampling and aggregation operations on the neighbor nodes of each node and obtain its own characteristic information by adding the current self-information and then updating it. We assume an undirected graph $Z=\left(V,E\right)$ with N nodes, as shown in Figure 2. The features of different nodes, as well as their updated features, are distinguished by different colors.

Here, ***V***
$\in R^{N\times F}$ is the node matrix in the graph. ***N*** represents the number of nodes, and *F* represents the dimension of each node feature in the graph. ***E***$\in R^{N\times N}$ is the adjacency matrix and represents the connection relationship between nodes. If there is an edge between two nodes denoted by *p* and *q*, $E_{p,q}$ = 1; otherwise, it is equal to 0. The set of nodes around a node with edges connected to it is called the set of neighbor nodes. When v is a node, $\varkappa \left(v\right)$ represents the set of neighbor nodes. The update process of GraphSAGE can be defined as follows [28]:(13)$$h_{\varkappa \left(v\right)}^{l}={AGGREGATE}_{l}(\{h_{u}^{l-1},\forall u\in \varkappa \left(v\right)\})\;,$$
(14)$$h_{v}^{l}=g(W^{l}CONCAT(h_{v}^{l-1},h_{\varkappa \left(v\right)}^{l})),$$
where $h_{\varkappa \left(v\right)}^{l}$ denotes the characteristics of neighbor nodes after aggregation. $h_{u}^{l-1}$ and $h_{v}^{l}$ represent the node feature information of layer l−1 and updated layer l, respectively. AGGREGATE is the aggregator using the averaging method for aggregation operations. CONCAT is the feature splicing operation, and g( · ) is an activation function.

For the non-uniform sparse array goniometry problem, the GraphSAGE convolutional layer updates the received signal characteristics of each group of elements by aggregating the information between different array groups. The addition of self-loop information can effectively fill the feature information at the vacant array elements and can update the features by aggregating the neighbor nodes and its own information. Therefore, the GNN has a certain degree of predictability for unknown information and can better adapt to the angle measurement situation under the conditions of random sparse arrays, low SNR, and small numbers of snapshots. Meanwhile, GNNs usually have strong generalization ability and can cope with radar goniometry tasks in different environments without the need for complex feature engineering in advance.

#### 2.2.2. GNN-DOA Model for Random Sparse Arrays

The covariance matrix contains the angular information of the target and can be adopted as a feature of the input neural network. Since each covariance matrix corresponds to a set of angle estimation results, this study constructs the DOA estimation as a graph-level classification problem. 

The DOA estimation model with a graph neural network (GNN) is shown in Figure 3. Each input covariance matrix is a graph and the covariance matrix is divided into a subset composed of multiple elements according to rows. Since the number of subsets equals the number of input nodes of the graph convolutional network, it is necessary to combine the received array signals in each subset into a set of vectors as the input features of the nodes. Finally, the adjacency matrix ***E*** is constructed according to the distance between each subset.

The backbone network consists of a feedforward neural network (FNN), a Graph Conv module with four SAGEConv layers and a global pooling layer. The graph convolution structure used here is GraphSAGE. It adds a self-loop to the adjacency matrix, can effectively aggregate the information between nodes and updates the embedded expression of its own nodes. 

When assuming z as the feature of input nodes, through two fully connected layers in FNN, the network output futures are $z_{1}$ and $z_{2}$. By introducing the weights of $w_{1}$ and $w_{2}$ and the biases of $b_{1}$ and $b_{2}$, the update process can be expressed as follows:(15)$$z_{1}=SiLU\left({w_{1}}^{T}z+b_{1}\right),$$
(16)$$z_{2}=SiLU\left({w_{2}}^{T}z_{1}+b_{2}\right),$$
where
(17)SiLU(a)=aϑ(a),
(18)$$\vartheta \left(b\right)=\frac{1}{1+e^{-b\;}},$$

The activation function SiLU(·) [32] nonlinearizes the output of the network.

Features enter the graph convolutional neural network for nodes’ feature updates. As shown in Figure 3, the SAGEConv layer consists of the GraphSAGE, the Layer Norm (LN) and an activation layer. The LN calculates the mean and variance of a layer of inputs and normalizes the data, which can stabilize the forward input distribution and accelerate convergence on mini-batch inputs. 

The update process of GraphSAGE is based on Equations (13) and (14). For simplicity, we define the Graph Conv module as a function G( · ). Therefore, the update process for the Graph Conv module is:(19)$$z_{3}=G\left(z_{2}\right),$$

After the node feature update is completed, the node features on each graph are mapped into a full graph representation through a global average pooling layer. The graph representation of the obtained feature $z_{3}$ is $H=\{h_{1},h_{2},h_{3},\ldots ,h_{N}\}$, and the feature representations of each node at each layer are aggregated using the common summation method of global pooling to obtain the graph-level representation of the feature.
(20)$$R(H)=\frac{1}{T}\sum_{i=1}^{N}h_{i},$$
where R( · ) is a commonly used readout function.

Finally, the graph-level features are classified through the fully connected layer. The network outputs the probability value of the presence of a target for each airspace interval, ranging from 0 to 1, and the closer the result is to 1, the greater the probability of the presence of a target in that direction.

#### 2.2.3. Outputs of the GNN Model

Generally, since the number of measured target signals is limited and sparse in airspace, the target airspace range can be evenly divided into *L* intervals, as shown in Figure 4. We can obtain $\varphi_{0}<\varphi_{1}<\cdots <\varphi_{L}$. If the target $\theta_{k}$ (*k* = 1, 2, …, *K*) exists in a sub-interval, the value of the interval section takes the value 1; otherwise, it takes a value of 0. The above concept can be described as the following formula:(21)$$f=\left\{\begin{array}{c} \;1,\;\;\varphi_{l-1}<\varphi_{k}<\varphi_{l}\\ \;0,\;\;\mathrm{otherwise} \end{array}\right.\;\;,\forall k\in \left\{1,2,\ldots ,K\right\};l=1,2,\ldots ,L.$$

Therefore, the DOA estimation constructs a sparse mapping relationship from the covariance matrix to the target airspace.

The specific algorithm steps are summarized as follows: The first step is to preprocess the received signal. The second step is to build graphs and nodes to build a graph convolutional neural network model. The third step is to test various arrays with different structures under different conditions. The fourth step is to optimize the network structure and achieve the best DOA estimation algorithm.

## 3. Simulation and Discussion

Table 1 lists the parameters of the training dataset. A random sparse linear array of a 24-element aperture has a sparsity of 0.5. That means 12 elements are randomly missing inside the array. Every six rows of the normalized covariance matrix are selected as a set of node features, and the number of nodes becomes four. To construct the adjacency matrix, we connect an edge between adjacent nodes with a distance of less than 2. In addition, the number of snapshots per sample is 50, and the SNR is randomly generated within the range of 0~10 dB. The discrete target space of −60°~+60° has a resolution of 1°, and 121 angle labels are obtained.

According to the above settings, the training dataset of two incoherent signals with 50,000 samples is generated. The training and testing datasets in the simulation are generated by MATLAB R2022a. The simulated samples are generated according to Equations (1)–(12), and the corresponding labels are generated according to Equation (21). 

The batch size and the learning rate used for training are 32 and 0.001. All neural network-based models were trained for 500 epochs. The optimizer is Adam, and the activation function is SiLU. We used ZLPR Loss [33] as the loss function for adjusting and updating parameters during training.

ZLPR Loss compares all non-target class scores with the target class scores, in order to achieve the effect that the target class score is greater than the score of each non-target class. The specific formula is provided as follows:(22)$$Loss=\log\left(1+\sum_{i\in \Omega_{neg}}e^{s_{i}}\right)+\log\left(1+\sum_{i\in \Omega_{pos}}e^{{-s}_{i}}\right),$$
where $\Omega_{neg}$ is the non-target class set, $\Omega_{pos}$ is the target class set and *S_i_* is the score.

When the result is greater than the threshold of 0.8, it is considered that there is a target in this direction, and the output result is set to 1; otherwise, it is set to 0. The accuracy calculation of the performance evaluation index is defined as follows: The total number of sample sets used for testing is recorded as TOTAL. If a sample is considered to be positive, the values of predictions and labels must be identical in all 121 dimensions. Meanwhile, the remaining samples are negative. The number of positive samples is recorded as TRUE, and the accuracy is calculated as follows:(23)$$Accuracy=\frac{TRUE}{TOTAL}\;,$$

In addition, mean absolute error (MAE) and root mean square error (RMSE) are two indicators used to evaluate DOA estimation error. MAE reflects the true error of DOA, while RMSE is more sensitive to outliers and can well reflect the robustness of DOA estimators. They are evaluated as follows: (24)$$MAE=\frac{1}{m}\sum_{i=1}^{m}\left|y_{true}-y_{pred}\right|,$$
(25)$$RMSE=\sqrt{\frac{1}{m}\sum_{i=1}^{m}{{(y}_{true}-y_{pred})}^{2}},$$
where *m* is the number of test samples. $y_{true}$ and $y_{pred}$ represent the true angle and pred angle, respectively. 

We now consider three experiments for analysis. Firstly, the proposed GNN algorithm is tested to estimate the performance from the angles of different sparsity arrays and compared with other algorithms. Secondly, the goniometric performance of GNN and traditional algorithms under the condition of a single snapshot is tested. Thirdly, the performance of GNNs for sparse arrays is examined. Finally, we compare the goniometric times of different algorithms, applying an Intel^®^ Core (TM) i7-9750Hv CPU @ 2.6 GHz hardware platform, based on CUDA 11.8 under the Pytorch framework.

### 3.1. Effect of the Array Sparsity on the Performance of DOA Estimation

In this experiment, to explore the performance of different angle estimation algorithms under different sparsity conditions, we select the SNR as 10 dB and the number of snapshots as 200. The traditional super-resolution algorithm MUSIC is chosen as a standard for comparison, while the existing neural network-based algorithm CNN and the DNN with MLP structure are compared. 

First of all, the RMSE values of different algorithms under different sparsity conditions are tested. As shown in Figure 5, the proposed GNN algorithm has excellent performance, even when the array loses one-third of its elements. Its RMSE remains as low as 0.04°, which is much lower than 13.12° for MUSIC and 0.79° for DNN. Since MLP cannot exchange information between different array elements, under the condition of sparse arrays, a large amount of information can indeed lead to algorithm performance degradation. On the one hand, the CNN uses local perception to extract feature information, and the connection between global features is not tight. On the other hand, the graph convolution module transmits updates layer by layer through the connection relationship between nodes. It effectively captures and models the spatial relationship between sensors. More specifically, the GNN algorithm uses the signals received by the adjacent antennas to update the characteristics of the nodes. It obtains global information through the aggregation and update of each layer. So, the GNN algorithm can build a complex nonlinear mapping relationship even if there are a large number of elements lost.

Meanwhile, the accuracies of different algorithms under different sparsity conditions are tested. It can be seen from Table 2 that when the array loses one-half of its elements, the neural network-based algorithms CNN and DNN are significantly better than the traditional goniometric algorithm MUSIC. This is because neural network models are data-driven and can learn more robust nonlinear representations through backpropagation mechanisms to combat noise and fluctuations in complex environments. The proposed algorithm guarantees an accuracy of 95.3% in this case, which proves that the GNN algorithm can effectively cope with the situation of losing a large number of array elements.

### 3.2. Effect of SNR and the Number of Snapshots on the Performance of DOA Estimation

In this experiment, under the conditions of an array sparsity of 1/3, the dependences of different algorithms on different SNR and numbers of snapshots are explored. We compare the GNN algorithm with MUSIC, the improved front-and-back smooth MUSIC (ssMUISC) [13], the neural network-based algorithms DNN and CNN and two excellent neural network generalization models ViT [34] and MLP-Mixer [35]. It is observed in Figure 6a that under the condition of 50 snapshots, all algorithms improved with an increase in SNR, among which the algorithm based on the neural network improved significantly, and the GNN model achieved the best performance above −6 dB. As shown in Figure 6b, under the condition of a signal-to-noise ratio of 10 dB, the number of snapshots is between 20 and 100, and the algorithm is not highly sensitive to its changes. This means that excellent angular performance can be achieved with a small number of snapshots. Even though MUSIC’s performance is poor in low-snapshot conditions, both GNN and ViT guarantee a very low RMSE with similarity. At the same time, the parameter size of the ViT model reproduced in the experiment is 1.18 M, while the proposed GNN model parameter is 690.93 KB, indicating that the GNN algorithm achieves the same high-precision performance as ViT with less computing resources. It is obvious that the proposed GNN model can reduce the computational burden without sacrificing performance and the Graph Conv structure used in the GNN algorithm can well adapt to the angle measurement conditions of sparse arrays with limited snapshots.

### 3.3. Effect of Single Snapshot and Large Array Sparseness on the Performance of DOA Estimation

Single-snapshot angle estimation algorithms are commonly used in engineering, and their calculation speed can quickly meet the needs of low-latency scenarios such as automatic driving. In this experiment, with a sparsity of 1/3, a single snapshot is used for angle measurement to explore the performance of different algorithms. Since DBF [8] is a very efficient single-snapshot angle estimation algorithm that can be implemented, the proposed algorithm is compared with it. As shown in Figure 7, MUSIC is difficult to cope with the situation of single-snapshot angle measurement when the array is sparse. With the improvement of SNR, MUSIC’s performance is not significantly improved, because it is difficult to solve the feature vectors corresponding to the signal space and noise space from the covariance matrix of a single snapshot. Under the condition that the array has a large sparsity, the RMSE of the DBF algorithm remains high. However, The GNN algorithm proposed can adapt well to the situation of a single snapshot, and the RMSE is significantly reduced with the improvement of SNR. It should be noted that the GNN algorithm is very robust when the array is sparse.

Furthermore, we tested the MAE of the single-snapshot algorithms on different sparsity arrays including RSLA12, RSLA16, RSLA20 and ULA. It can be seen from Figure 8 that when the SNR is higher than 5 dB, the GNN algorithm combined with RSLA16, RSLA20 and ULA achieves better angular measurement performance than the DBF algorithm applied to the ULA. Therefore, the proposed algorithm can adopt fewer antenna array elements occupying the same antenna aperture and obtain a level of performance only achieved by the traditional algorithm applied to the ULA. It also contributes to reducing the manufacturing cost of radar antennas. Especially in practical applications, a sparse array configuration, which can effectively reduce the coupling between physical antennas, has important engineering value.

Finally, we compare the traditional DOA estimation algorithms with the proposed algorithm based on a graph neural network. The test experiment calculates the average of 1000 Monte Carlo experiments on an Intel^®^ Core (TM) i7-9750Hv CPU @ 2.6GHz hardware platform. It can be found in Figure 9 that the time calculation cost of the GNN algorithm is nearly half that of MUSIC and is slightly higher than that of the DBF algorithm. It is obvious that the proposed algorithm can achieve better angular measurement performance while ensuring low computational cost and is more in line with the real-time requirements of engineering angle estimation algorithms.

## 4. Conclusions

This work proposes a GNN-based DOA estimation algorithm. The graph convolutional network structure is introduced into the field of DOA estimation for achieving robust angle estimation on non-uniform random sparse arrays. The GNN model can effectively fill in the information at the missing elements by aggregating and updating the characteristics of neighbor nodes in the convolutional structure and provides good robustness in the angle estimation problem of sparse arrays. In the experiments, the proposed algorithm achieves better goniometric performance, compared with the traditional algorithm and the existing deep neural network algorithm based on multilayer perceptron and convolutional structure in a complex environment. We tested the proposed algorithm on an array with a sparsity of 1/3. When the SNR is −4 dB and the number of snapshots is 50, the angle measurement performance is 88.81% higher than that of MUSIC. In the case of limited snapshots or even a single snapshot, the proposed GNN algorithm has no dependence on the specific array arrangement and is well adapted to arrays with a large number of random missing elements. In addition, the array sparseness can effectively reduce the manufacturing cost of antennas and reduce the coupling between elements. Meanwhile, the GNN algorithm has a low time calculation cost and can meet the requirements of scenarios requiring low-latency angle estimation. This proves the advanced and efficient introduction of graph convolutional networks into DOA estimation problems. 

In the future, we will further investigate the DOA of sparse arrays for incoherent sources, study the DOA of virtual sparse arrays and improve the robustness of the model. In addition, experimental verification by measured data will be conducted soon.

## Figures and Tables

**Figure 1 sensors-24-00091-f001:**
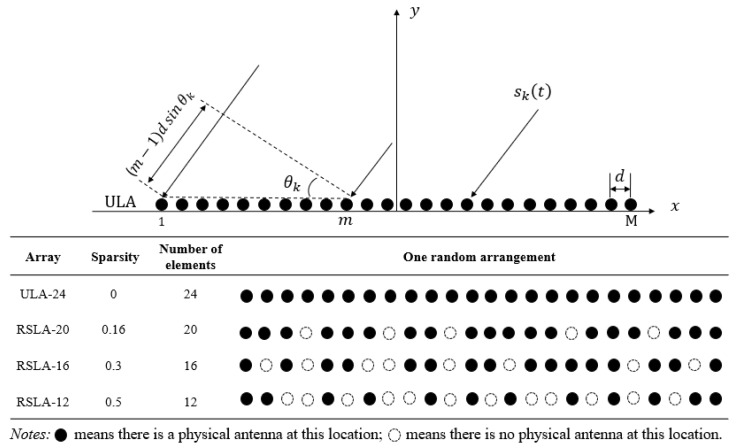
Twenty-four-element aperture uniform linear array and random sparse linear array structure.

**Figure 2 sensors-24-00091-f002:**
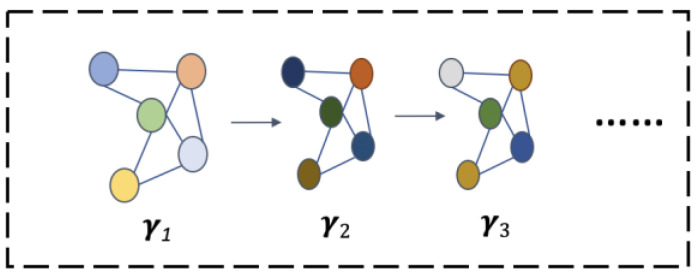
Graph structure and its update process.

**Figure 3 sensors-24-00091-f003:**
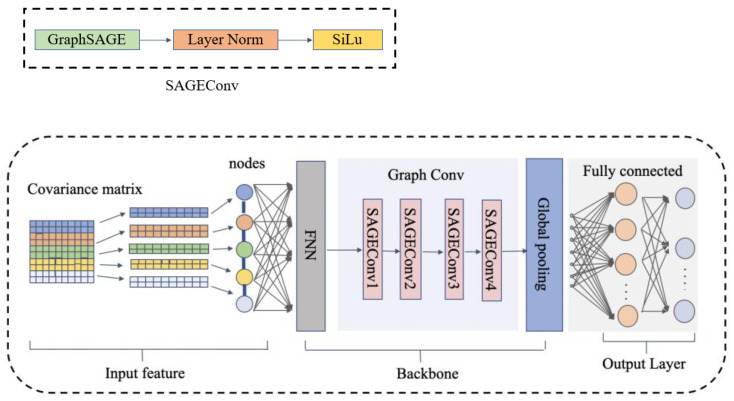
DOA estimation model with GNN.

**Figure 4 sensors-24-00091-f004:**
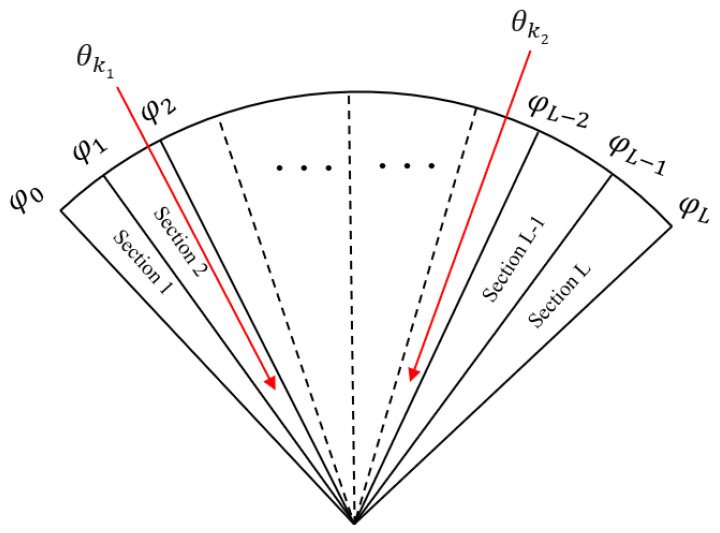
The relationship between target airspace and sections of GNN labels.

**Figure 5 sensors-24-00091-f005:**
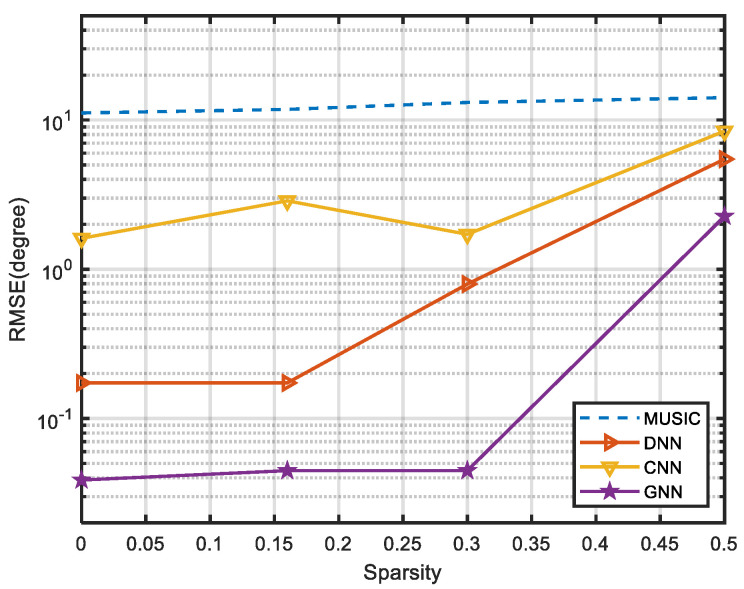
RMSE of different algorithms under different conditions of sparsity.

**Figure 6 sensors-24-00091-f006:**
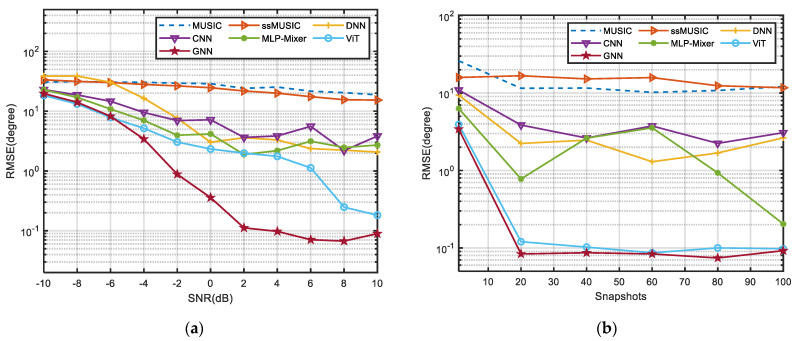
(**a**) RMSE of different algorithms with different SNR. (**b**) RMSE of different algorithms with different snapshots.

**Figure 7 sensors-24-00091-f007:**
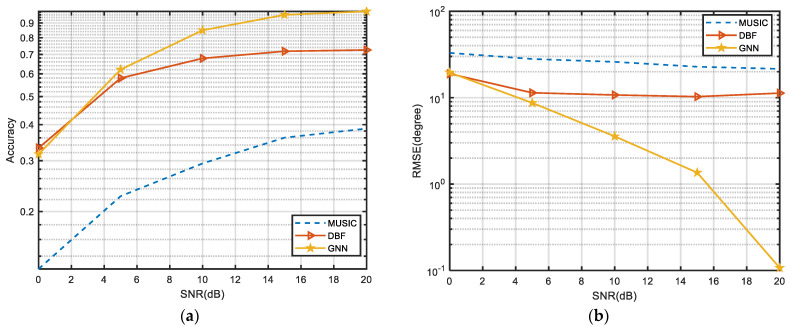
(**a**) RMSE of different algorithms under the single-snapshot condition. (**b**) Accuracy of different algorithms under the single-snapshot condition.

**Figure 8 sensors-24-00091-f008:**
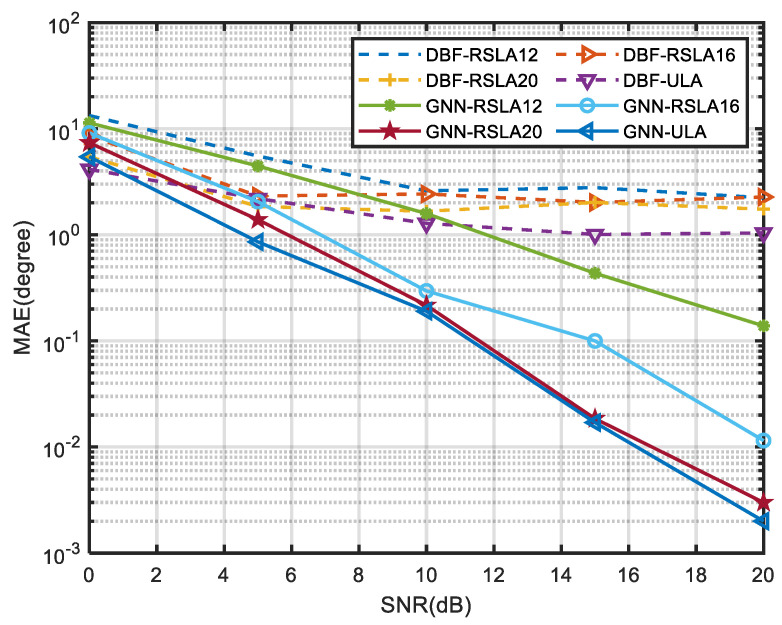
MAE of DBF and GNN with different SNR on arrays of differing sparsity.

**Figure 9 sensors-24-00091-f009:**
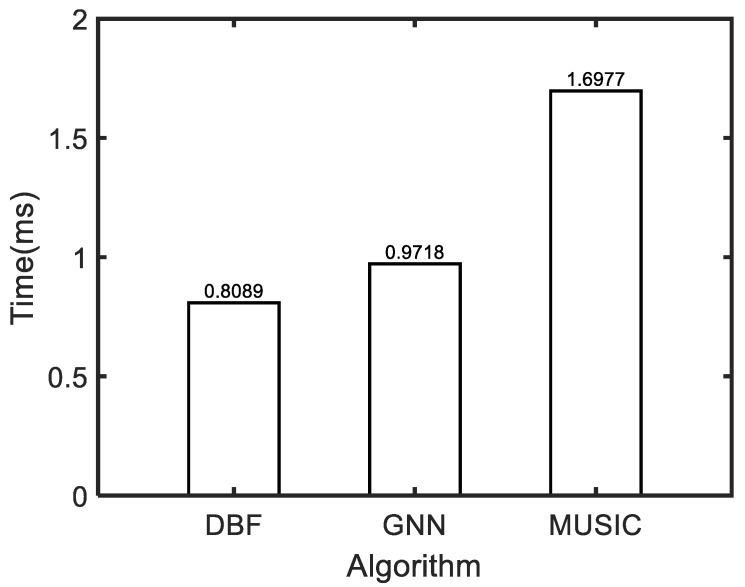
Comparison of running time of different DOA estimation algorithms.

**Table 1 sensors-24-00091-t001:** Training parameters and values.

Training Parameters	Values
Number of array sensors	12
Sparsity of the array	0.5
Number of incoherent sources	2
Angular range	[−60°–60°]
Number of sections	121
Angular resolution	1°
SNR	0–10 dB
Number of snapshots	50

**Table 2 sensors-24-00091-t002:** Accuracy of different algorithms under different sparsity conditions.

Number of Antennas	12	16	20	24
MUSIC	58.5%	68.8%	79.4%	83.6%
DNN	81.3%	92.9%	97.6%	98.4%
CNN	87.1%	97.1%	99.1%	99.0%
GNN	95.3%	99.3%	99.8%	99.8%

## Data Availability

The data presented in this study are available on request from the corresponding author. The training dataset used in this article is obtained through simulation and is not a public dataset.

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
