# Peer review of "Direction-of-Arrival Estimation for a Random Sparse Linear Array Based on a Graph Neural Network"

_sensors, 2023, doi:10.3390/s24010091_

Round 1

Reviewer 1 Report

Comments and Suggestions for Authors

The manuscript “DOA Estimation for a Random Sparse Linear Array based on a Graph Neural Network” by Yiye Yang et al. proposes an interesting array signal processing problem for the estimation of direction-of-arrival (DOA) for a random sparse linear array.

The authors claim that their algorithm achieves high-precision DOA estimation and strong robustness in complex environments, such as low SNR states, low snapshots, defective arrays, missing elements, or a combination of these situations based on the prejudice that the GNN can update the feature information of each element by aggregating the received signals between neighboring elements, and can restore to some extent the element information that is randomly missing in the non-uniform array.

However, the creative imaginations of the authors could not be scientifically validated. The manuscript needs significant improvement. The explicit methodology of this manuscript is not demonstrated concerning the key claim of the authors as ascribed in the above para. There are no demonstrations and results for the method to address complex environments, such as low SNR states, low snapshots, defective arrays, missing elements, or a combination of these situations, etc. and hence has not achieved anything better or is not evident in the manuscript compared to the other traditional methods wherein such complex situations are demonstrated by authors worldwide.

The connection between each element and the selection of subsets or nodes is not described in detail in the manuscript. The scheme given in the figures neither explains that from an array element level. This connection must address complex environments, such as low SNR states, low snapshots, defective arrays, missing elements, or a combination of these situations, etc. The frequency band and the range of targets are not discussed. The theoretical assumptions given in the manuscript do not consider the dynamics associated.

There is no experimental validation either.

Comments on the Quality of English Language

NIL

Reviewer 2 Report

Comments and Suggestions for Authors

1. The paper uses GNN for the direction-of-arrival (DOA) estimation algorithm for a random sparse linear array. paper can be improved by first providing significantly more detailed information on the background of the estimation algorithm.

2. GNN's performance depends on how the graphs are constructed. The paper doesn't explain how the edge relationship between nodes are being established (I am assuming nodes are individual antenna in the array).

3. How does your algorithm compare with the 'Maximum-Likelihood Method' from the paper https://arxiv.org/pdf/2203.13433.pdf

4. Since it is a machine-learning method, what are you hoping to achieve once a machine-learning trained model has been obtained after training on the data?  Also, please first explain the dataset used for the paper. How are you hoping to deploy the trained machine learning model to make inferences?

Reviewer 3 Report

Comments and Suggestions for Authors

Manuscript (sensors-2692485): DOA Estimation for a Random Sparse Linear Array based on a Graph Neural Network.

In this manuscript, the authors proposed the direction-of-arrival (DOA) estimation algorithm for a random sparse linear array based on a novel graph neural network (GNN).

However, the major weakness of the manuscript is its limited contribution.

Besides, this reviewer has also some major concerns regarding the assumptions detailed below.

1) Insert Figure 2 after mentioned in the text.

2) Authors are needed to clear the paper contributions in abstract section. 

3) In Table 2, the unit of data is missed. 

4) Mention the future work in "Conclusion section".  

5) In this manuscript, a lot of mathematically equations are used but not mention references. All are equations derived by you. 

Round 2

Reviewer 1 Report

Comments and Suggestions for Authors

The authors have revised their manuscript quite satisfactorily. It can be considered for publication in its present form.

Comments on the Quality of English Language

Nil

Author Response

Thank you for your valuable comments on the improvements to this manuscript.

Reviewer 3 Report

Comments and Suggestions for Authors

Thank you for addressing my all concerns......

Author Response

All the authors are very grateful for your professional advice.